# MITIGATING UNOBSERVED CONFOUNDING VIA DIFFUSION PROBABILISTIC MODELS

## ABSTRACT

Learning Conditional average treatment effect estimation from observational data is a challenging task due to the existence of unobserved confounders. Previous methods mostly focus on assuming the Ignorability assumption ignoring the unobserved confounders or overlooking the impact of an apriori knowledge on the generation process of the latent variable, which can be quite impractical in real-world scenarios. Motivated by the recent advances in the latent variable modeling, we propose to capture the unobserved latent space using diffusion model, and accordingly to estimate the causal effect. More concretely, we build on the reverse diffusion process for the unobserved confounders as a Markov chain conditioned on an apriori knowledge. In order to implement our model in a feasible way, we derive the variational bound in closed form. In the experiments, we compare our model with the state-of-the-art methods based on both synthetic and real-world datasets, demonstrating consistent improvements of our model.

## 1 INTRODUCTION

Estimating the Conditional average treatment effect estimation (CATE) or Heterogeneous Treatment Effect (HTE) from observational data is a fundamental problem across a wide variety of domains. For example, re-weighting the training instances with the inverse propensity score (IPS) in recommender system (Wang et al., 2021; 2022), measuring the effect of a certain medicine against a disease in healthcare (Shalit, 2020) and providing counterfactual visual explanations in computer vision (Goyal et al., 2019). In this paper, we focus on these measure problems from confounders perspective.

How to measure the confounder is an essential problem in estimating CATE of an treatment $A$ (e.g.,medicine) on an individual with features $X$ (e.g., demographic characteristics ). A confounder is a variable which affects both the treatment and the outcome. On the one hand, one can account for CATE by controlling it with the Ignorability assumption in mind, i.e., there does not exists the unobserved confounder. The most crucial mechanism lie in balance the distribution among groups, usually through inverse propensity weighting (IPW) or covariate adjustment (Yao et al., 2021; Louizos et al., 2017). While quite a lot of promising models have been proposed and achieved impressive performance, such as, the representative CFR (Shalit et al., 2017), the augmented IPW estimator DR (Funk et al., 2011) and so on, these methods build on the Ignorability assumption, which can be impractical in real-world scenarios. On the other hand, exactly collecting all of valid confounders is impossible in the general case. For example, demographic characteristics and genetic factor can both affect the choice of medication to a patient, and the patient's health. However we can only have access to the former in the observational data. As illustrated in Figure 1, the genetic factor acts as an unobserved confounder $Z$ both affecting the treatment $A$ and health outcomes $Y$, and without controlling it we can not block the backdoor path: $A \leftarrow Z \rightarrow Y$ as of estimating the causal effect of treatments on health measures.

In the past few years, some prominent generative models have been proposed to generate such unobserved confounder that we could utilize it to isolate the causal effect of treatment on outcome. For instance, VAE-based method CEVAE (Louizos et al., 2017) assume that there exists a proxy variable in causal graph, and then generates the hidden confounder $Z$ by optimizing the variational lower bound of this graphical model, GANITE (Yoon et al., 2018) aims to generate the counterfactual distributions using GAN, and accordingly to infer the CATE in an unbiased settings. Additionally, some advanced techniques are also applied to reconstruct or generate the hidden confounder, like

Gaussian Processes (Witty et al., 2020), Imitation Learning (Zhang et al., 2020), deep latent variable models (Josse et al., 2020) and more (Li & Zhu, 2022; Yao et al., 2021).

While great success has been made, these methods have some intrinsic limitations for modeling hidden confounder. Since the latent confounder distribution does not follow a specific form in the real-world system, the main challenge is that GAN and VAE-based cannot fully describe the latent information. Besides the main challenge, GAN-based methods could be unstable in modeling CATE due to the adversarial losses. VAEs make substantially weaker assumptions in generating the structure of the hidden confounders (Louizos et al., 2017), which could restrict the model's flexibility.

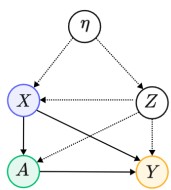

Table 1: Motivating example on the generation process of the unobserved confounders. $\eta$ represents the common prior variable of observed confounders $X$ and unobserved confounders $Z$. $A$ and $Y$ denote the treatment and outcome respectively.

In order to address these challenges, in this paper, we propose to generate the unobserved confounders using diffusion model. We aim to exploit two types of generation process called forward diffusion process and reverse diffusion process, respectively. The former process converts the observed confounders to a simple noise distribution by adding noise at each time step, in which the useful decomposition information can be preserved in transition kernel. And then the transition kernel are utilized to the unobserved confounder generation. The latter process are regarded as a Markov chain which is responsible for converting the noise distribution to our target latent distribution. By integrating these two processes, we can learn its transition kernel and accordingly reconstruct the desired unobserved confounders. Furthermore, we also design a generation factor as the condition for learning the transition kernel. The generation factor follows a prior distribution in our setting of generation. As illustrated in Figure 1, we assume that the generation factor $\eta$ can simultaneously affect the generation process of the observed confounder and the unobserved confounder. For examples, the environment in which the patient live and work can both affect the patient clinical data and gene for a certain disease. Therefore, the environment can be regarded as a generation factor, which plays a significant role in generating unobserved confounders. Moreover, the clinical data could be affected by the patient gene, and hence which can be used to generate the unobserved information, like gene.

The main contributions of this paper can be concluded as follows: (1) We propose to solve the task of unobserved confounders in causal inference with the diffusion model. (2) To realize the above idea, we first derive a variational lower bound of the likelihoodof the unobserved confounders conditional on the generation factor, and then reformulate that bound into a tractable expression in closed form. (3) We verify the effectiveness and generality of our framework by comparing with 12 state-of-the art methods on four datasets. The empirical studies manifest that the proposed method can achieve competitive gains both on synthetic and benchmark datasets.

## 2 Preliminaries

In the context of estimating CATE, understanding the underlying mechanisms of data generation and transformation is crucial. The Diffusion Denoising Probabilistic Model (DDPM) framework, which simulates the conversion of real data into Gaussian noise and its subsequent reversal, provides a powerful tool for modeling complex data distributions. This background is essential for our application of diffusion models to identify unobserved confounders—a critical step in accurately estimating CATE under the presence of hidden biases.

### 2.1 Estimation of Treatment Effect

The Conditional Average Treatment Effect (CATE), is defined as:

$$\tau(x) := \mathbb{E}[Y_1 - Y_0 \mid x]$$

where $Y_a$ represents the potential outcome under treatment $a$, and $x$ denotes the covariates or characteristics of the individual. This measure quantifies the expected difference in outcomes when

the treatment is applied versus when it is not, conditioned on the individual's characteristics. For more details, see the Appendix.

## 2.2 Diffusion Model

DDPMs simulate the data generation process by reversing a diffusion process that transforms real data $\boldsymbol{x}^0$ into Gaussian noise $\boldsymbol{x}^T$ over time (Ho et al., 2020). The process $p_\theta(\boldsymbol{x}^0)$ is defined as:

$$p_\theta(\boldsymbol{x}^0) = \int p(\boldsymbol{x}^T) \prod_{t=1}^{T} p_\theta(\boldsymbol{x}^{t-1} \mid \boldsymbol{x}^t) \, d\boldsymbol{x}^{1:T}$$

The sequence $\boldsymbol{x}^{T:0}$ is defined as a Markov chain with learned Gaussian transitions, each denoted by:

$$p_\theta(\boldsymbol{x}^{t-1} \mid \boldsymbol{x}^t) = \mathcal{N}(\mu_\theta(\boldsymbol{x}^t, t), \Sigma_\theta(\boldsymbol{x}^t, t)) \tag{1}$$

This formulation shows how the model uses parameterized Gaussian transitions to reverse the diffusion process step-by-step, recreating the initial data from pure noise.

**Forward Process (Diffusion).** In the forward process, starting with the data sample $\boldsymbol{x}^0$ from the distribution $q(\boldsymbol{x}^0)$, noise is incrementally added over $T$ time steps, until the data is completely converted into Gaussian noise $\boldsymbol{x}^T$. The noise addition at each step $t$ is defined by:

$$q\left(\boldsymbol{x}^{(t)} \mid \boldsymbol{x}^{(t-1)}\right) = \mathcal{N}\left(\boldsymbol{x}^{(t)}; \sqrt{\bar{\alpha}_t}\boldsymbol{x}^{(0)}, (1 - \bar{\alpha}_t)\boldsymbol{I}\right) \tag{2}$$

where $\bar{\alpha}^t = \prod_{i=1}^{t} \alpha^i$, and $\alpha^t = 1 - \beta^t$ represents how much of the previous data is retained (with $\beta^t \in (0, 1)$ is a hyper-parameter ). The $\alpha^t$ terms are crucial as they determine the rate at which the data is corrupted by noise.

**Reverse Process (Denoising).** Recall that we use Equation 1 to denoise. Typically, the mean is calculated using the expression derived by the reparameterization trick and Bayes' rule:

$$\mu_\theta(\boldsymbol{x}^t, t) = \frac{1}{\sqrt{\alpha^t}} \left( \boldsymbol{x}^t - \frac{\beta^t}{\sqrt{1 - \bar{\alpha}^t}} \boldsymbol{\epsilon}_\theta(\boldsymbol{x}^t, t) \right),$$

where $\bar{\alpha}^t = \prod_{i=1}^{t} \alpha^i$. In this process, $\boldsymbol{\epsilon}_\theta(\boldsymbol{x}^t, t)$ represents the noise estimated by the parameterized network. This equation facilitates the step-by-step transformation from pure noise back to structured data. The covariance matrix is typically fixed to $\beta^t \boldsymbol{I}$ in practice.

**Learning the Noise Model.** The training of DDPM involves learning the function $\boldsymbol{\epsilon}_\theta$ that can accurately predict the noise $\boldsymbol{\epsilon}$ added at each step based on the noisy data $\boldsymbol{x}^t$ and the step number $t$. The loss function used typically minimizes the mean squared error between the actual noise and the predicted noise:

$$\mathcal{L}(\theta) = \mathbb{E}_{\boldsymbol{x}^0, \boldsymbol{\epsilon}, t} \left[ \|\boldsymbol{\epsilon} - \boldsymbol{\epsilon}_\theta(\sqrt{\bar{\alpha}^t}\boldsymbol{x}^0 + \sqrt{1 - \bar{\alpha}^t}\boldsymbol{\epsilon}, t)\|^2 \right], \quad \text{where } \boldsymbol{\epsilon} \sim \mathcal{N}(0, I).$$

The loss function encourages the model to accurately infer the noise components that were added to the data, allowing the reverse process to effectively denoise the data.

**Intuition** Diffusion models uniquely decompose the data generation process into "denoising" steps, progressively transforming noise into complete samples (Letafati et al., 2023). This approach distinctly sets diffusion models apart from other generative models like GANs (Gui et al., 2021) and VAEs (Kingma & Welling, 2019), which do not feature a denoising step in their generative process.

## 3 Diffusion Model for unobserved confounders

In this section, we develop a diffusion model to identify unobserved confounders. We then reformulate our training objective using a closed-form variational bound for efficient model training. Finally, we illustrate how the generated unobserved features are utilized to improve the training process and estimation of CATE.

---

**Algorithm 1** Inference of Unobserved Confounders

---

**Input:** Observed data point $\boldsymbol{x}$.
Calculate the posterior $q_{\boldsymbol{\varphi}}(\boldsymbol{\eta} \mid \boldsymbol{x})$;
Sample data points $\boldsymbol{z}^{(T)} \sim \mathcal{N}(0, I)$;
Use the learned reverse process to estimate $p_{\boldsymbol{\theta}}(\boldsymbol{z}^{(t-1)} \mid \boldsymbol{z}^{(t)}, \boldsymbol{\eta})$ **for** $t = T, T-1, \ldots, 1$;
**Return:** The unobserved confounders $\boldsymbol{z}^{(0)}$.

---

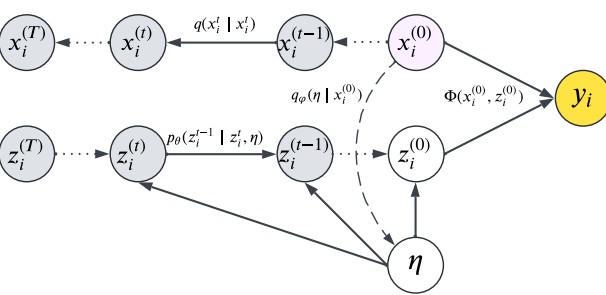

Figure 1: Corresponding graph model of our method: The sequence from $\boldsymbol{x}_i^{(0)}$ to $\boldsymbol{x}_i^{(T)}$ illustrates the forward diffusion process, whereas the sequence from $\boldsymbol{z}_i^{(T)}$ to $\boldsymbol{z}_i^{(0)}$ illustrates the reverse denoising process. The variable $\boldsymbol{\eta}$ represents the common prior for the variables $X$ and $Z$, thereby linking their distributions. $Y$ denotes the outcome variable.

## 3.1 INFERENCE UNOBSERVED CONFOUNDERS USING DIFFUSION MODEL

The process of conditional image generation using diffusion models has been extensively explored (Luo & Hu, 2021; Zhang et al., 2023; Ni et al., 2023). Unlike the well-documented generation of images where generated outputs can be directly compared with training data, the generation of unobserved confounders $\boldsymbol{Z}$ presents unique challenges due to the absence of observable data for $\boldsymbol{Z}$. This issue necessitates the development of effective representations for unobserved variables. We address the challenge of learning these representations in Section 3.2. The formulation begins with using observed data $\boldsymbol{X}$ to infer unobserved confounders $\boldsymbol{Z}$ through a diffusion model.

Recall that our goal is to generate unobserved confounders $\boldsymbol{Z}$, as shown in Figure 1. We propose introducing a latent variable $\boldsymbol{\eta}$ to capture the shared parental information of the observed variable $\boldsymbol{X}$ and the unobserved variable $\boldsymbol{Z}$. Subsequently, we can use the observed data $\boldsymbol{x}$ to infer the posterior $p(\boldsymbol{\eta} \mid \boldsymbol{x}^{(0)})$ and then generate the corresponding $\boldsymbol{z}$ using the likelihood $p(\boldsymbol{z} \mid \boldsymbol{\eta}, x^{(T)})$.

The forward diffusion process in our task involves incrementally adding noise to the observed variable $\boldsymbol{x}^{(0)}$, transforming the initial distribution into a pure noise distribution This transformation occurs incrementally over $T$ steps, culminating in $\boldsymbol{x}^{(T)}$. This procedure adheres to the standard diffusion process outlined in Section 2.2.

In our generation process, the reverse diffusion is capable of approximating $p_{\boldsymbol{\theta}}(\boldsymbol{z}^{(0)} | \boldsymbol{z}^{(1)}, \boldsymbol{\eta})$ from a simple noise distribution $p_{\theta}(\boldsymbol{z}_i^{(T)})$ that are given as the input. For the inference process, the rationale for deriving $\boldsymbol{z}^{(0)}$ from $\boldsymbol{x}^{(0)}$ is that, for example, the fundamental characteristics of a patient can reveal underlying factors, such as environmental traits, among others. Therefore, with the latent representation $\boldsymbol{\eta}$ and the preserved information from forward diffusion process, we can generate the desired unobserved confounders $\boldsymbol{z}$ through the reverse Markov chain. Formally, the reverse diffusion process for generating unobserved confounders is:

$$p_{\boldsymbol{\theta}}(\boldsymbol{z}^{(0:T)}|\boldsymbol{\eta}) = p(\boldsymbol{z}^{(T)}) \prod_{t=1}^{T} p_{\boldsymbol{\theta}}(\boldsymbol{z}^{(t-1)}|\boldsymbol{z}^{(t)}, \boldsymbol{\eta}) \tag{3}$$

where $p_{\boldsymbol{\theta}}(\boldsymbol{z}^{(t-1)}|\boldsymbol{z}^{(t)}, \boldsymbol{\eta})$ is learnable transition kernel and $\boldsymbol{\theta}$ is the model parameters. It describes the denoising process at some time steps. The learnable transition kernel takes the form of

$$p_{\boldsymbol{\theta}}(\boldsymbol{z}^{(t-1)}|\boldsymbol{z}^{(t)}, \boldsymbol{\eta}) = \mathcal{N}(\boldsymbol{z}^{(t-1)}; \mu_{\boldsymbol{\theta}}(\boldsymbol{z}^{(t)}, t, \boldsymbol{\eta}), \beta_t \boldsymbol{I}) \tag{4}$$

In this model, the mean $\mu_{\boldsymbol{\theta}}(\boldsymbol{x}^{(t)}, t, \boldsymbol{\eta})$ are parameterized by deep neural networks learned in the optimization process and $\boldsymbol{\eta}$ is the latent representation encoding the generation factor. Unlike the setup described in Section 2.2, the additional variable $\boldsymbol{\eta}$ establishes the dependence between $\boldsymbol{x}$ and $\boldsymbol{z}$, facilitating the inference of the posterior $q_{\boldsymbol{\varphi}}(\boldsymbol{\eta} \mid \boldsymbol{x}^{(0)})$ and enabling the sampling of unobserved variable $\boldsymbol{Z}$ accordingly.

In practice, we assume the noise distribution $p(\boldsymbol{x}_i^{(T)})$ to be a standard normal distribution $\mathcal{N}(0, \boldsymbol{I})$. By applying the reverse Markov chain which given the generation factors and initial distribution $p(\boldsymbol{x}_i^{(T)})$, we can retrieve the unobserved confounders aligned with the target distribution.

**Inference of Unobserved Confounders.** With above well-defined denoising process established, we can now apply it to causal inference. As depicted in Algorithm 1 and Figure 1, each time we observe a data point $\boldsymbol{x}$, the process starts by calculating the posterior $q_{\boldsymbol{\varphi}}(\boldsymbol{\eta} \mid \boldsymbol{x}^{(0)})$, which models the latent representation $\boldsymbol{\eta}$ given the observed data. Subsequently, the algorithm samples a point $\boldsymbol{z}^{(T)}$ from a standard normal distribution $\mathcal{N}(0, I)$, initializing the reverse diffusion sequence. This sampled data point serves as the basis for the reverse diffusion process, which iteratively estimates $\boldsymbol{z}^{(t-1)}$ from $\boldsymbol{z}^{(t)}$ using the transition kernel $p_{\boldsymbol{\theta}}$ conditioned on $\boldsymbol{\eta}$. This iterative process proceeds until $t = 1$, finally yielding the inferred unobserved confounders $\boldsymbol{z}^{(0)}$. These confounders, alongside the initial observation $\boldsymbol{x}$, allow the model to predict the potential outcomes $y_i$ as outlined in the Figure 1. The model thus leverages both observed and latent variables to generate comprehensive predictions that integrate both observed characteristics and inferred unobserved factors.

**Variational Lower Bound.** With the formulated forward and reverse diffusion processes for unobserved confounders in mind, we now aims to formalize the training objective. Since directly optimizing the exact log-likelihood is intractable, we instead maximize its variational lower bound (VLB)(the detailed derivation is present in the Appendix):

$$\mathbb{E}[-\log p_{\boldsymbol{\theta}}(\boldsymbol{z}^{(0)})] \leq \underbrace{E_q \left[ \log \frac{q(\boldsymbol{x}^{(1:T)}, \boldsymbol{\eta}|\boldsymbol{x}^{(0)})}{p_{\boldsymbol{\theta}}(\boldsymbol{z}^{(0:T)}, \boldsymbol{\eta}))} \right]}_{VLB} \tag{5}$$

Where $L_{VLB}$ is a common objective for training probabilistic generative models (Luo & Hu, 2021; Ho et al., 2020; Yang et al., 2023). The intuition behind the lower bound on the right-hand side (RHS) is rooted in the fact that both the unobserved and observed variables share the same prior. Given the absence of labels for the unobserved variable $Z$, we assume a similar distribution for these variables and utilize KL divergence to control the training process, a common strategy in representation learning (Louizos et al., 2017; Schölkopf et al., 2021). Additionally, this approach offers the benefit of stabilizing the training process, as opposed to relying solely on regression errors on the potential outcome $Y$. Further exploration of the loss function design will be discussed in Section 3.2.

We can further derive the $L_{VLB}$ as:

$$L_{VLB} = E_q \left[ \log \frac{q(\boldsymbol{x}^{(1:T)}, \boldsymbol{\eta}|\boldsymbol{x}^{(0)})}{p_{\boldsymbol{\theta}}(\boldsymbol{z}^{(0:T)}, \boldsymbol{\eta}))} \right]$$

$$= E_q \left[ \sum_{t=2}^{T} D_{KL} \left( \underbrace{q(\boldsymbol{x}^{(t-1)}|\boldsymbol{x}^{(t)}, \boldsymbol{x}^{(0)})}_{A} || \underbrace{p_{\boldsymbol{\theta}}(\boldsymbol{z}^{(t-1)}|\boldsymbol{z}^{(t)}, \boldsymbol{\eta})}_{B} \right) \right.$$

$$\left. - \log \underbrace{p_{\boldsymbol{\theta}}(\boldsymbol{z}^{(0)}|\boldsymbol{z}^{(1)}, \boldsymbol{\eta})}_{C} + D_{KL} \left( \underbrace{q_{\boldsymbol{\varphi}}(\boldsymbol{\eta}|\boldsymbol{x}^{(0)})}_{D} || \underbrace{p(\boldsymbol{\eta})}_{E} \right) \right] \tag{6}$$

The above training objective can be optimized efficiently since each term in this objective is tractable. In order to make the objective more clear, we elaborate on the terms as following:

A $q(\boldsymbol{x}^{(t-1)}|\boldsymbol{x}^{(t)}, \boldsymbol{x}^{(0)})$ is computed by a closed-form Gaussian (Luo & Hu, 2021; Ho et al., 2020):

$$q(\boldsymbol{x}^{(t-1)}|\boldsymbol{x}^{(t)}, \boldsymbol{x}^{(0)}) = \mathcal{N}(\boldsymbol{x}^{(t-1)}; \boldsymbol{\mu}_t(\boldsymbol{x}^{(t)}, \boldsymbol{x}^{(0)}), \gamma_t \boldsymbol{I}) \tag{7}$$

Where $\boldsymbol{\mu}_t(\boldsymbol{x}^{(t)}, \boldsymbol{x}^{(0)}) = \frac{\sqrt{\bar{a}_{t-1}}\beta_t}{1-\bar{a}_t}\boldsymbol{x}^{(0)} + \frac{\sqrt{a_t}(1-\bar{a}_{t-1})}{1-\bar{a}_t}\boldsymbol{x}^{(t)}$ and $\gamma_t = \frac{1-\bar{a}_{t-1}}{1-\bar{a}_t}\beta_t$.

B, C $p_{\boldsymbol{\theta}}(\boldsymbol{z}^{(t-1)}|\boldsymbol{z}^{(t)}, \boldsymbol{\eta})$ where $t \in \{1, 2, ..., T\}$ are trainable Gaussian distribution shown in Eq. 4. D $q_{\boldsymbol{\varphi}}(\boldsymbol{\eta}|\boldsymbol{x}^{(0)})$ are learnable posterior distribution, which is the posterior of $\boldsymbol{\eta}$ after observe $\boldsymbol{x}^{(0)}$, aiming to encode the input observed confounders $\boldsymbol{x}^{(0)}$ into the distribution of the latent generation factor $\boldsymbol{\eta}$. We define it as: $q_{\boldsymbol{\varphi}}(\boldsymbol{\eta}|\boldsymbol{x}^{(0)}) = \mathcal{N}(\boldsymbol{\eta}; \boldsymbol{\mu}_{\boldsymbol{\varphi}}(\boldsymbol{x}^{(0)}), \sum_{\boldsymbol{\varphi}}(\boldsymbol{x}^{(0)}))$. The last term E, $p(\boldsymbol{\eta})$ is the prior distribution defined as isotropic Gaussian $\mathcal{N}(0, \boldsymbol{I})$, which is the most common choice for approximating the target distribution. In the next section, we will show how to optimize this objective for generating the desired unobserved confounders $\boldsymbol{Z}$ and accordingly estimating CATE.

## 3.2 ALGORITHM FOR ESTIMATING CATE

As introduced in the previous section, the process of inferring unobserved confounders involves calculating the posterior of latent variables from observed data, sampling from a noise distribution, and iteratively applying a reverse diffusion process to estimate and retrieve the unobserved confounders at each step. Utilizing these generated unobserved features, we enhance the regression model to improve the estimation of the CATE. We then address the remaining question of how to learn effective unobserved representations in Section 3.1.

Let $\Phi : \mathcal{X} \to \mathcal{R}$ be a representation function, $f : \mathcal{R} \times \{0, 1\} \to \mathcal{Y}$ be an hypothesis predicting the outcome of a patient's confounders $x$ given the representation confounders $\Phi(\boldsymbol{x})$ and the treatment assignment $a$. Let $L : \mathcal{Y} \times \mathcal{Y} \to \mathbb{R}_+$ be a loss function. The estimation of CATE by an hypothesis $f$ and a representation function $\Phi$ is :

$$\hat{\tau}(\boldsymbol{x}) = f(\Phi(\boldsymbol{x}), 1) - f(\Phi(\boldsymbol{x}), 0) \tag{8}$$

We utilize the expected Precision in Estimation of Heterogeneous Effect (PEHE) (Hill, 2011) to train our model. We define it as following:

$$\epsilon_{PEHE}(f) = \int_{\mathcal{X}} (\hat{\tau}(\boldsymbol{x}) - \tau(\boldsymbol{x}))^2 p(\boldsymbol{x}) dx \tag{9}$$

Following above analysis, we propose a method called DFHTE ( Estimation of **H**eterogeneous **T**reatment **E**ffect Using **DiF**fusion Model), which take into account the unobserved confounders to estimate CATE.

The optimization problem in our framework is shown as the following:

$$\min_{f, \Phi, \theta, \varphi} \sum_{i=1}^{m} w_i \cdot L(y_i, f(\Phi(\boldsymbol{x_i}, \boldsymbol{z_i}), a_i)) + L_{VLB}(\boldsymbol{x_i}) + \alpha \cdot \text{IPM}_G(\hat{p}_{\Phi}^{a=1}, \hat{p}_{\Phi}^{a=0}) \tag{10}$$

where $w_i$ is used to compensates for the difference in treatment group size. It can be calculated be the proportion of treated units in the population, the loss funcation $L$ is PEHE. the unobserved confounder $z_i$ is derived by diffusion model, i.e., $\boldsymbol{z_i} \sim \mu_{\boldsymbol{\theta}}(\boldsymbol{c}, t, \eta_i) + \beta_t \epsilon$ where $\epsilon, c \sim \mathcal{N}(0, I)$, $t$ is the time step in reverse Markov chain and $q_{\varphi}(\eta_i|\boldsymbol{x_i})$ is the generation factor. Note that in this setup, the representation function $\Phi$ takes as input both observed and generated features, $\boldsymbol{x_i}$ and $\boldsymbol{z_i}$ respectively. Here, we use reparameterization trick to make the generation process feasible. $L_{VLB}$ is the VLB loss that aims to learn the transition kernel. In practice, optimizing $L_{VLB}$ in our main objective is still a challenging task, since it requires to sum the expectation of the KL terms on all time steps. To make the training more efficient, we adopt the works in Ho et al. (2020) randomly choosing one term to optimize at each training step. The detailed training algorithm is present in Appendix. $\hat{p}_{\Phi}^{t=1}$ and $\hat{p}_{\Phi}^{t=0}$ are learned high-dimensional representation for treated and control groups respectively, $\text{IPM}_G(\cdot, \cdot)$ is the (empirical) integral probability metric w.r.t. a function family $G$. We adopt it to balance the treated and control distribution. The imbalance penalty $\alpha$ are used to weight the magnitude of the two distribution.

Building upon the optimization methodologies discussed earlier, our approach generates latent confounders influenced by a generative factor derived from a noise distribution. This method not only facilitates a precise estimation of the CATE but also utilizes regression loss to assess the quality of the generated features.

We refer to the model minimizing equation 10 with the observed and unobserved confounders as DFHTE. The model are trained by the adaptive moment estimation (Adam) (Kingma & Ba, 2014). The details are described in the Appendix.

Table 2: Conditional average treatment effect estimation on ACIC, IHDP and two types of Sim datasets. The top module consists of baselines from recent works. The bottom module consists of our proposed method. In each module, we present each of the result with form mean ± standard deviation and we use bold fonts to label the best performance. Lower is better.

| Datasets | ACIC | | IHDP | | Sim-$z$ | | Sim-$\eta$ | |
|---|---|---|---|---|---|---|---|---|
| Metric | $\sqrt{\epsilon_{PEHE}}$ | $\epsilon_{ATE}$ | $\sqrt{\epsilon_{PEHE}}$ | $\epsilon_{ATE}$ | $\sqrt{\epsilon_{PEHE}}$ | $\epsilon_{ATE}$ | $\sqrt{\epsilon_{PEHE}}$ | $\epsilon_{ATE}$ |
| RF | 3.09 ± 1.48 | 1.16 ± 1.40 | 4.61 ± 6.56 | 0.70 ± 1.50 | 4.92 ± 0.00 | 0.61 ± 0.01 | 12.13 ± 0.00 | 3.21 ± 0.02 |
| CF | 1.86 ± 0.73 | 0.28 ± 0.27 | 4.46 ± 6.53 | 0.81 ± 1.36 | 4.70 ± 0.00 | 0.74 ± 0.00 | 6.96 ± 0.00 | 1.25 ± 0.00 |
| S-learner | 3.86 ± 1.45 | 0.41 ± 0.35 | 5.76 ± 8.11 | 0.96 ± 1.80 | 4.96 ± 0.00 | 0.84 ± 0.00 | 11.74 ± 0.00 | 0.92 ± 0.00 |
| T-learner | 2.33 ± 0.86 | 0.79 ± 0.68 | 4.38 ± 7.85 | 2.16 ± 6.17 | 5.68 ± 0.08 | 0.94 ± 0.10 | 6.87 ± 0.12 | 1.05 ± 0.29 |
| CEVAE | 5.63 ± 1.58 | 3.96 ± 1.37 | 7.87 ± 7.41 | 4.39 ± 1.63 | 5.20 ± 0.03 | 1.78 ± 0.12 | 12.83 ± 0.61 | 5.37 ± 0.47 |
| BNN | 2.00 ± 0.86 | 0.43 ± 0.36 | 3.17 ± 3.72 | 1.14 ± 1.70 | 5.09 ± 0.04 | 1.37 ± 0.19 | 12.49 ± 0.21 | 5.04 ± 0.52 |
| DragonNet | 1.26 ± 0.32 | 0.15 ± 0.13 | 1.46 ± 1.52 | 0.28 ± 0.35 | 4.09 ± 0.10 | 0.50 ± 0.32 | **6.16 ± 0.10** | 0.47 ± 0.30 |
| TARNet | 1.30 ± 0.46 | **0.15 ± 0.12** | 1.49 ± 1.56 | 0.29 ± 0.40 | 4.10 ± 0.11 | 0.52 ± 0.34 | **6.16 ± 0.10** | 0.44 ± 0.36 |
| GANITE | 4.27 ± 1.34 | 3.27 ± 1.37 | 6.79 ± 5.60 | 4.43 ± 1.43 | 4.07 ± 0.06 | 1.92 ± 0.09 | 10.78 ± 0.15 | 5.83 ± 0.20 |
| CFR$_{MMD}$ | 1.24 ± 0.31 | 0.17 ± 0.14 | 1.51 ± 1.66 | 0.30 ± 0.52 | 4.06 ± 0.09 | **0.40 ± 0.32** | 6.16 ± 0.11 | 0.45 ± 0.33 |
| CFR$_{WASS}$ | 1.27 ± 0.38 | 0.15 ± 0.12 | 1.43 ± 1.61 | 0.27 ± 0.41 | 4.10 ± 0.09 | 0.52 ± 0.36 | 6.18 ± 0.11 | 0.49 ± 0.35 |
| QHTE | 1.32 ± 0.41 | 0.19 ± 0.18 | 1.83 ± 1.90 | 0.34 ± 0.43 | 6.05 ± 0.23 | 0.58 ± 0.26 | 7.39 ± 0.38 | 0.84 ± 0.43 |
| DFHTE | **1.20 ± 0.07** | 0.20 ± 0.14 | **0.59 ± 0.08** | **0.17 ± 0.11** | **4.05 ± 0.08** | 0.41 ± 0.3 | 6.17 ± 0.12 | **0.44 ± 0.34** |

In this section, we outline the development of a diffusion model tailored for identifying and generating unobserved confounders, which includes detailing both the forward and reverse diffusion processes. We introduce a variational lower bound formulation as our training objective, which facilitates efficient model optimization by approximating the intractable log-likelihood. Lastly, we demonstrate how these synthesized unobserved confounders are integrated into regression models to enhance the accuracy of CATE estimation.

## 4 EXPERIMENTS

### 4.1 EXPERIMENT SETUP

This section outlines our experimental approach for assessing the effectiveness of the proposed DFHTE model in estimating CATE across a variety of datasets. We conduct experiments using two synthetic datasets, Sim-$z$ and Sim-$\eta$, designed to mimic scenarios with unobserved confounders, and two benchmark datasets, ACIC 2016 (Dorie et al., 2019) and IHDP (Hill, 2011), which are commonly used in causal inference research. These datasets provide a comprehensive test bed due to their varied complexity and the nature of the confounders involved. For detailed descriptions of the datasets and experimental setup, please see the appendix. Additionally, DFHTE's performance is compared against a wide array of established causal inference models, ensuring a thorough validation of its capabilities in diverse scenarios. We adopt the commonly used metrics including Rooted Precision in Estimation of Heterogeneous Effect (PEHE) (Hill, 2011) and Mean Absolute Error (ATE) (Shalit et al., 2017) for evaluating the quality of CATE. Formally, they are defined as: $\sqrt{\epsilon_{PEHE}} = \sqrt{\frac{1}{n}\sum_{i=1}^{n}(\hat{\tau}_i - \tau_i)^2}, \epsilon_{ATE} = |\frac{1}{n}\sum_{i=1}^{n}(\hat{\tau}) - \frac{1}{n}\sum_{i=1}^{n}(\tau)|$, where $\hat{\tau}_i$ and $\tau_i$ stand for the predicted CATE and the ground truth CATE for the $i$-th instance respectively. The more details about the implementation of all adopted baselines and our methods and full experimental settings are presented in following Appendix.

### 4.2 OVERALL RESULTS

The overall comparison results are presented in Table 2, from which we can see: among the baselines, distance metric methods like CFR$_{WASS}$ and CFR$_{MMD}$, can obtain more performance gain both than the non-distance metric ones like GANITE and CEVAE, and traditional machine learning models like RF and CF, in most cases. This observation is consistent to our expectations and also agrees with the previous work (Shalit et al., 2017), and verify that minimizing the distance between the treated

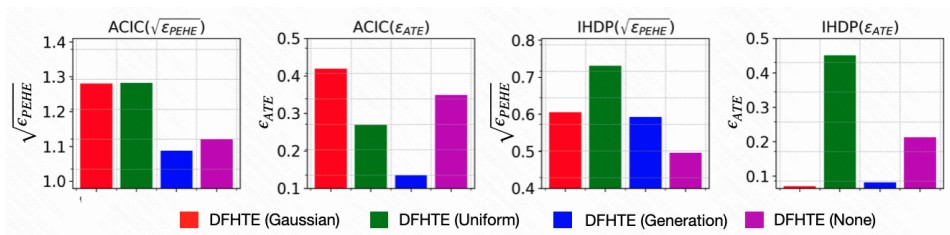

Figure 2: Performance comparison between our model and its variants on the unobserved confounders. The performances of different types of unobserved confounders are labeled with different colors. Lower is better.

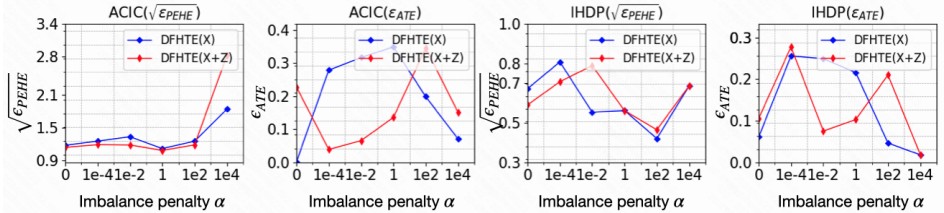

Figure 3: Influence of the imbalance penalty $\alpha$ on our model performance in terms of $\sqrt{\epsilon_{PEHE}}$ and $\epsilon_{ATE}$. The performances of different types of confounders are labeled with different colors. Lower is better.

and control groups on the studied latent space can effectively eliminate the distribution shift and lead to better performance on CATE estimation.

It is encouraging to see that our model DFHTE can achieve the best performance on different datasets and evaluation metrics in more cases. The results verify the effectiveness of our idea. Comparing with the baselines, we take advantages of both the observed and unobserved confounders, which enable us to not only facilitate the identification of potential outcome, but also enhance to balance the studied representations between the treated and control groups. As a result, our model can always achieve the better performance on the estimation of CATE.

### 4.3 CONFOUNDERS CERTIFICATION

In this section, we would like to study whether different unobserved confounders in our model are necessary. To this end, we compare our model with four different unobserved confounders: DFHTE(Gaussian) is a method with the unobserved confounders sampled randomly from the normal Gaussian $\mathcal{N}(0,1)$, DFHTE(Uniform) is based on Uniform $\mathcal{U}(-1.5, 1.5)$, DFHTE(Generation) is our method, in which the unobserved confounder are generated by a reverse diffusion model and DFHTE(None) is the typical representation methods with the ignorability assumption hold. Due to the space limitation, we present the results based on $\sqrt{\epsilon_{PEHE}}$ and $\epsilon_{ATE}$ and the datasets of ACIC and IHDP. From the results shown in Figure 2, we can see: DFHTE(Gaussian) performs better than DFHTE(Uniform). We speculate that the unobserved confounders sampled from normal Gaussian is more common than sampled from Uniform in practice. Nevertheless, both of which performs worse than DFHTE(None). This maybe because by randomly drawing unprovable unobserved confounders, the CATE model are forced to encode the the noise samples, which result in a biased estimation. It is interesting to see that when we add the generated confounders in estimating CATE, the performance of DFHTE(Generation) is better than DFHTE(None) in more cases. This observation demonstrates the effectiveness of our idea on capturing the unobserved confounders.

### 4.4 PARAMETER STUDY

In this section, we analyze the influence of the key hyper parameters in our objective 10, we report the results on the same datasets and evaluation metric as the above experiments. The imbalance penalty $\alpha$ determines the magnitude of IPM in overall loss function. We tune $\alpha$ in $[0, 1e-4, 1e-2, 1, 1e2, 1e4]$. In order to investigate the influence of the unobserved confounders in parameter study, we compare our model with its two combinations of confounders: DFHTE(X) is a model based on the observed

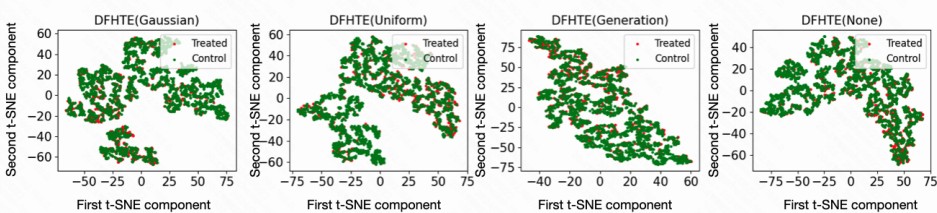

Figure 4: t-SNE visualization of the balanced representations of ACIC learned by our algorithm DFHTE with 4 types of unobserved confounders.

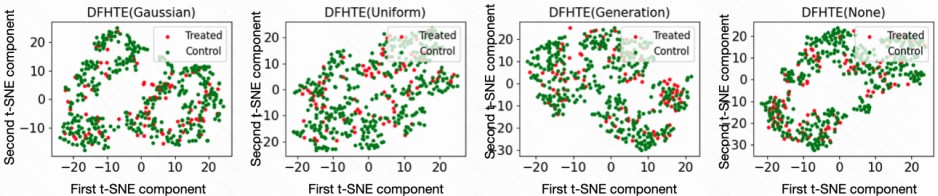

Figure 5: t-SNE visualization of the balanced representations of IHDP learned by our algorithm DFHTE with 4 types of unobserved confounders.

confounder X and DFHTE(X+Z) is based on both the observed and unobserved confounders X and Z, where $Z$ is generated by a reverse diffusion model. The results are presented in Figure 3, from which we can see: for both methods of DFHTE(X) and DFHTE(X+Z), the performance fluctuates a lot as $\alpha$ varies, but the best performance is usually achieved when $\alpha$ is moderate. This agrees with our expectation, i.e., too small $\alpha$ may lead to the imbalanced studied representation, while too large $\alpha$ may hinder the accurate estimation of CATE. Between DFHTE(X) and DFHTE(X+Z), we can find that the red line usually appears below blue line. The intuitive example suggests that the performance of DFHTE(X+Z) tend to better than DFHTE(X) as $\alpha$ varies. As expected, the unobserved confounders generated by our methods contributes to the estimation of CATE and should not be ignored.

## 4.5 LEARNED REPRESENTATIONS

In this section, we investigate the influence of different types of unobserved confounders in balancing the studied representations between treated and control groups, where the parameter settings follow the above experiments, and we compare the explanations generated by DFHTE(Gaussian), DFHTE(Uniform),DFHTE(Generation) and DFHTE(None). From the results shown in Figure 4 and Figure 5 we can see: all of these methods can perform several regions where the representations are indeed balanced. Such that they appear equal in studied high-dimension space. The results demonstrate that the distance metric used to balance two distributions play a significant role in improving the estimation of CATE. Furthermore, in the illustration of representations generated by DFHTE(Generation), we can find that some regions appear a strip-like representation on IHDP, whereas some regions appear rod-like shape on ACIC, where both of which have a smaller overlap. This observation demonstrate that the unobserved confounders generated by reverse diffusion model can contribute to balancing the studied distribution between treated and control groups.

## 5 CONCLUSION

In this paper, we propose to generate the unobserved confounders, and accordingly to facilitate the identification of potential outcome, as well as enhancing the learned representations. To achieve this goal, we first reconstruct the unobserved confounders by a reverse diffusion model, and then to estimation the CATE and balance the distribution between the treated and control groups based on the combination of the observed and unobserved confounders. In the experiments, we evaluate our framework based on both synthetic and real-world datasets to demonstrate its effectiveness and generality. This paper makes a first step on applying the idea of diffusion model to the field of estimating CATE. There is still much room for improvement. To begin with, one can incorporate different prior knowledge into the generation process, and at the same time devise effective mechanism for encouraging identification to causal inference. In addition, in order to reduce the time-consuming, people can also investigate the specific time step in generating the unobserved confounders.

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

## A  BROADER IMPACTS

Our methods is first work that enables diffusion models on generating the unobserved confouners, which could help to break the back-door path in measuring the treatment effects. We have demonstrated that by using diffusion model we can improve the accurate of treatment effects. We believe that the proposed method could facilitate the research community. Additionally, there is no ethics problems in generating process. The reason is that the generated unobserved factors are consist of numeral vectors.

## B  RELATED WORK

**Treatment Effect Estimation.** Accurate and correct estimation of Conditional average treatment effect estimation is an challenging task in real-world scenarios, since the lack of counterfactuals can lead to an biased estimation from observational study. To alleviate this problem, early methods, like re-weighting models (Austin, 2011; Imai & Ratkovic, 2014; Fong et al., 2018), use the Inverse propensity weighting (IPW) mechanism to reduce selection bias based on covariates. Another active top line of research is to incorporate traditional machine learning into the study of estimating CATE, like Bayesian Additive Regression Trees (BART) (Hill, 2011), Random Forests (RF) (Breiman, 2001), Causal Forests (CF) (Wager & Athey, 2018), etc. In order to balance the distribution among groups in representation space, some advanced models are designed, like DragonNet (Shi et al., 2019) , CFR (Shalit et al., 2017), QHTE (Qin et al., 2021), etc. There models use more flexibility and sophisticated technique, like Integral Probability Metric (IPM), to pull in that distributions and while minimize generalization bound for CATE estimation. While remarkable progresses have made by these models, the premise that they need to get the Ignorability assumption hold.  However, the Ignorability assumption is untestable in practice. To this end, some promising deep generative models are proposed to generate latent variables. For example, Causal Effect Variational Autoencoder (CEVAE) (Louizos et al., 2017) leverage Variational Autoencoders to obtain the unobserved confounders and simultaneously infer causal effects, GAITE (Yoon et al., 2018) use Generative Adversarial Nets (GANs) framework to capture the uncertainty in the counterfactual distributions. While remarkable progresses have made by these models, here are some intrinsic limitations for modeling latent variables. For examples, GAN-based methods could be unstable in modeling CATE due to the adversarial losses. VAEs make substantially weaker assumptions in generating the structure of the hidden confounders (Louizos et al., 2017), which could restrict the model's flexibility.  In this paper, we build on diffusion model to generate the unobserved confounders and accordingly to measure CATE. The benefits are presented in two aspects; (1) Comparing to CEVAE, diffusion model has less assumptions in our settings, which is great of importance for estimating CATE; (2) Diffusion model has a comparative stable loss function, which indeed contribute to the generation process of unobserved confounders.

**Diffusion Model.** Diffusion Model is a concept describing the study of the deep generative process. It basically involves two types of Markov chains, called forward diffusion process and reverse diffusion process respectively.  The former is capable of converting any data distribution into a simple or noise prior distribution, while the latter aims to reconstruct the original data distribution by a reverse Markov chain. In that process, the goal is to learn a transition kernels parameterized by deep neural networks (Yang et al., 2022) and accordingly to generate the desired data. Due to its flexibility and strength, recent years have witnessed many studies on incorporating diffusion model into a variety of challenging domains (Yang et al., 2022; Luo & Hu, 2021; Ho et al., 2020) and achieved impressive results. For example, inspired by the diffusion model in computer vision,  Luo & Hu (2021) proposes to generate 3D point cloud by a Markov chain conditioned on certain shape latent. In natural language processing, in order to handle more complex controls in generating text, Diffusion-LM (Li et al., 2022) is proposed as a new language model based on continuous diffusion. Additionally, Adaptive Denoising Purification (Yoon et al., 2021) proposes an effective randomized purification scheme to purify attacked images in robust learning. Similar to these applications, in this paper, we proposed to generate the unobserved confounders by a Markov chain conditioned on the generate factor that is derived from the observed confounders. To the best of our knowledge, this is the first work on estimating conditional average treatment effect estimation.

## C  BACKGROUND: HETEROGENEOUS TREATMENT EFFECT

Under the Neyman-Rubin potential outcomes framework (Rubin, 2005), CATE estimation aims to measure the causal effect of a treatment or intervention $a \in \mathcal{A}$ on the outcome $y \in \mathcal{Y}$ for given the unit's confounders or descriptions $x \in \mathcal{X}$. Throughout this paper, we only focus on the binary treatment case, where $\mathcal{A} = \{0, 1\}$, $y$ represents the factual outcome. We treat units which received treatment, i.e., $a = 1$ as treated units and the other units with $a = 0$ as control units. The Conditional Average Treatment Effect (CATE) for unit $x$ is (Shalit et al., 2017):

$$\tau(x) := \mathbb{E}[Y_1 - Y_0 | x] \tag{11}$$

Where $Y_a$ denotes the potential outcome for treatment $a$. In practice, we can only observe the factual outcome with respect to treatment assignment, i.e., $y = Y_0$ if $a = 0$, otherwise $y = Y_1$. Usually, we build on three significant assumptions to guarantee that the potential outcomes are identifiable from observational study.

**Assumption 1.** *Consistency. For a given patient with treatment assignment a, then the potential outcome for the treatment a is the same as the observed (factual) outcome: $Y_a = y$*

**Assumption 2.** *Positivity (Overlap) . if $P(X = x) \neq 0$, then $P(A = a | X = x) > 0, \quad \forall a$ and $x$.*

**Assumption 3.** *Strong ignorability. For a given patient $(i)$, the treatment are independent of the potential outcomes if given the confounders $X : A \perp\!\!\!\perp Y_1, Y_0 | X$.*

With these assumptions in mind, the estimation on potential outcomes could be transformed into identifiable estimation from a statistical point of view. In other words, we call that the counterfactual outcomes can be identified under these assumptions, i.e, $\tau(x) = \mathbb{E}[Y | X = x, A = 1] - \mathbb{E}[Y | X = x, A = 0]$. From machine learning perspective, these observational dataset can be modeled via a standard supervised learning model, such as SVM, for estimating $\tau(x)$. However, this model could be unreliable and unviable employed to estimate the future counterfactual outcomes under the fact that without adjusting for the bias introduced by the unobserved confounders and imbalanced distribution between treated groups and control groups. The existing generative-based models can achieve promising results in generating unobserved confounders (Louizos et al., 2017) and counterfactuals (Yoon et al., 2018), which indeed eliminate the influence from backdoor between treatment and outcome. However, they have some inherent limitations, which would hinder the model's flexibility and performance. In this paper, we build on the prominent diffusion model to generate the unobserved confounders, and accordingly align the distribution between treated groups and control groups and measure the CATE. We proceed in two steps: (1) Generate the unobserved confounders conditioned on generation factor; (2) Balance the confounder's representation in latent space and measuring the CATE based on the observed and unobserved confounders.

## D  EXPERIMENT DETAILS.

### D.1  DATASETS AND SIMULATION

CATE estimation is more difficult compared to machine learning tasks, the reason is that we rarely have access to ground-truth treatment effect in real-world scenario. In order to measure the accurate estimation of CATE, we conduct experiments based on two types of synthetic datasets and two standard benchmark datasets. The detailed description about these datasets are shown as follows:

**ACIC 2016.** This is a common benchmark dataset introduced by Dorie et al. (2019), which was developed for the 2016 Atlantic Causal Inference Conference competition data. It comprises 4,802 units (28% treated, 72% control) and 82 confounders measuring aspects of the linked birth and infant death data (LBIDD). The dataset are generated randomly according to the data generating process setting. We conduct experiments over randomly picked 100 realizations with 63/27/10 train/validation/test splits.

**IHDP.** Hill (2011) introduced a semi-synthetic dataset for causal effect estimation. The dataset was based on the Infant Health and Development Program (IHDP), in which the confounders were generated by a randomized experiment investigating the effect of home visits by specialists on future cognitive scores. it consists of 747 units(19% treated, 81% control ) and 25 confounders measuring

Table 3: Statistics of the datasets used in our experiments.

| Dataset | #Replications | #Units | #confounders | Treated Ratio | Control Ratio |
|---------|---------------|--------|--------------|---------------|---------------|
| ACIC | 100 | 4,802 | 82 | 28% | 72% |
| IHDP | 1,000 | 747 | 25 | 19% | 81% |
| Sim-$z$ | 100 | 10,000 | 50 | 50% | 50% |
| Sim-$\eta$ | 100 | 10,000 | 50 | 50% | 50% |

the children and their mothers. Following the common settings in Qin et al. (2021); Shalit et al. (2017), We average over 1000 replications of the outcomes with 63/27/10 train/validation/test splits.

**Sim-$z$.** This synthetic dataset is based on observed and unobserved confounders that are both obtained from an normal Gaussian distribution. We adopt the generation process proposed in Assaad et al. (2021); Louizos et al. (2017) to simulate the treatment effect as:

$$
\begin{aligned}
&x_i \sim \mathcal{N}(0, \sigma_X^2); \;\; z_i \sim \mathcal{N}(0.5, \sigma_Z^2); \\
&a_i | x_i, z_i \sim \text{Bernoulli}(\sigma(0.5 x_i^T \beta_X + 0.5 z_i^T \beta_Z)) \\
&\epsilon_i \sim \mathcal{N}(0, \sigma_Y^2); \;\; \mathbf{y}_i(0) = x_i^T \beta_a + z_i^T \beta_b - r + \epsilon_i \\
&\mathbf{y}_i(1) = x_i^T \beta_a + z_i^T \beta_b + x_i^T \beta_c + z_i^T \beta_d + r + \epsilon_i
\end{aligned}
\tag{12}
$$

where $\sigma$ is the logistic sigmoid function. This generation process satisfies the assumptions of ignorability and positivity. We randomly construct 100 replications of such datasets with 10,000 units (50% treated, 50% control) and 50 confounders by setting $\sigma_X$ and $\sigma_Y$ both to 0.5, $\beta_T$, $\beta_0$ and $\beta_1$ are all sampled from $\mathcal{N}(0, 1)$.

**Sim-$\eta$.** This synthetic dataset aims to mimic the causal data generating process in terms of a prior distribution specified in advance. We simulate the treatment effect as:

$$
\begin{aligned}
&\boldsymbol{\eta}_i \sim \mathcal{N}(0, I); \\
&x_i | \boldsymbol{\eta} \sim \mathcal{N}(\eta_i, \sigma_{x_1}^2 \eta_i + \sigma_{x_0}^2 (1 - \eta_i)); \\
&z_i | \boldsymbol{\eta} \sim \mathcal{N}(\eta_i + 0.5, \sigma_{z_1}^2 \eta_i + \sigma_{z_0}^2 (1 - \eta_i));
\end{aligned}
\tag{13}
$$

We sample the generation factor $\eta$ from a standard normal distribution and accordingly generate the confounder $x$ and $z$. The remaining generation process is the same as Sim-$\eta$. We randomly construct 100 replications of such datasets with 10,000 units (50% treated, 50% control) and 50 confounders by setting $\sigma_{x_1}^2$, $\sigma_{x_0}^2$, $\sigma_{z_1}^2$, and $\sigma_{z_0}^2$ to 0.5,0.3,0.7 and 0.9 respectively.

The statistics of the datasets are presented in Table 3.

**Baselines.** We compare our model with the following 12 representative baselines: Random Forests (RF) (Breiman, 2001), Causal Forests (CF) (Wager & Athey, 2018), Causal Effect Variational Autoencoder (CEVAE) (Louizos et al., 2017), DragonNet (Shi et al., 2019), Meta-Learner algorithms S-Learner (Nie & Wager, 2021) and T-Learner (Künzel et al., 2019), Balancing Neural Network (BNN) (Johansson et al., 2016), Treatment-Agnostic Representation Network (TARNet) (Shalit et al., 2017), Estimation of Conditional average treatment effect using generative adversarial nets (GANITE) (Yoon et al., 2018) as well as CounterFactual Regression with the Wasserstein metric (CFR$_{WASS}$) (Shalit et al., 2017) and the squared linear MMD metric (CFR$_{MMD}$) (Shalit et al., 2017), along with a extension of CRF method Query-based Heterogeneous Treatment Effect estimation (QHTE) (Qin et al., 2021).

**Implementation details.** We implement our methods based on QHTE (Qin et al., 2021). We adopt the commonly used metrics including Rooted Precision in Estimation of Heterogeneous Effect (PEHE) (Hill, 2011) and Mean Absolute Error (ATE) (Shalit et al., 2017) for evaluating the quality of CATE. Formally, they are defined as:

$$
\sqrt{\epsilon_{PEHE}} = \sqrt{\frac{1}{n} \sum_{i=1}^{n} (\hat{\tau}_i - \tau_i)^2}, \; \epsilon_{ATE} = |\frac{1}{n} \sum_{i=1}^{n} (\hat{\tau}) - \frac{1}{n} \sum_{i=1}^{n} (\tau)|
\tag{14}
$$

---

**Algorithm 2** Training

---

Indicate the observational data $\mathcal{X}$.
Initialize all the model parameters.
**while** not converged  Sample $\boldsymbol{x}^{(0)} \sim \mathcal{X}$

Sample $\boldsymbol{\eta} \sim q_{\boldsymbol{\varphi}}(\boldsymbol{\eta}|\boldsymbol{x}^{(0)})$

Sample $t \sim \text{Uniform}(\{1, ..., T\})$

Sample $\boldsymbol{x}_1^{(t)}, ..., \boldsymbol{x}_m^{(t)} \sim q(x^{(t)}|x^{(0)})$

$$L_\theta = \sum_{i=1}^m D_{KL}\left(q(\boldsymbol{x}_i^{(t-1)}|\boldsymbol{x}^{(t)}, \boldsymbol{x}_i^{(0)})||p_{\boldsymbol{\theta}}(\boldsymbol{z}^{(t-1)}|\boldsymbol{z}_i^{(t)}, \boldsymbol{\eta})\right)$$

$$L_\varphi = D_{KL}\left(q_{\boldsymbol{\varphi}}(\boldsymbol{\eta}|\boldsymbol{x}^{(0)})||p(\boldsymbol{\eta})\right)$$

Compute the gradients of the $L_\theta + \frac{1}{T}L_\varphi$
Perform the gradient descent.

---

**Algorithm 3** Sampling

---

Sampling data points: $\boldsymbol{z}^{(T)} \sim \mathcal{N}(0, \boldsymbol{I})$.
**for** $t = T, ..., 1$ $\epsilon \sim \mathcal{N}(0, \boldsymbol{I})$ if $t > 0$, else $\epsilon = 0$
$\boldsymbol{z}^{(t-1)} = \mu_{\boldsymbol{\theta}}(\boldsymbol{z}^{(t)}, t, \eta) + \beta_t \epsilon$
return unobserved confounders $\boldsymbol{z}^{(0)}$

---

where $\hat{\tau}_i$ and $\tau_i$ stand for the predicted CATE and the ground truth CATE for the $i$-th instance respectively. The more details about the implementation of all adopted baselines and our methods and full experimental settings are presented in following Appendix.

### D.2 IMPLEMENTATION AND EVALUATION OF THE DFHTE MODEL

We implement our methods based on QHTE (Qin et al., 2021). We use the same set of hyperparameters for DFHTE across four datasets. More precisely, we employ 3 similar fully-connected exponential-linear layers for the encoder $q_{\boldsymbol{\varphi}}(\boldsymbol{\eta}|\boldsymbol{x}^{(0)})$, the transition kernel $p_{\boldsymbol{\theta}}(\boldsymbol{x}^{(t-1)}|\boldsymbol{x}^{(t)}, \boldsymbol{\eta})$, representation function $\Phi$, and the CATE prediction function $f$ respectively. The difference is that layer sizes are 128 for both $q_{\boldsymbol{\varphi}}(\boldsymbol{\eta}|\boldsymbol{x}^{(0)})$ and $p_{\boldsymbol{\theta}}(\boldsymbol{x}^{(t-1)}|\boldsymbol{x}^{(t)}, \boldsymbol{\eta})$, 200 for $\Phi$, and 100 for $f$. we use Batch normalization (Ioffe & Szegedy, 2015) to facilitate training, and all but the output layer use ReLU (Rectified Linear Unit) (Agarap, 2018) as activation functions. In the main optimization objective, we set $\alpha$ and $\beta$ both to 1. We adopt the commonly used metrics including Rooted Precision in Estimation of Heterogeneous Effect (PEHE) (Hill, 2011) and Mean Absolute Error (ATE) (Shalit et al., 2017) for evaluating the quality of CATE. Formally, they are defined as:

$$\sqrt{\epsilon_{PEHE}} = \sqrt{\frac{1}{n}\sum_{i=1}^n (\hat{\tau}_i - \tau_i)^2}, \quad \epsilon_{ATE} = |\frac{1}{n}\sum_{i=1}^n (\hat{\tau}) - \frac{1}{n}\sum_{i=1}^n (\tau)| \tag{15}$$

where $\hat{\tau}_i$ and $\tau_i$ stand for the predicted CATE and the ground truth CATE for the $i$-th instance respectively.

## E DETAILED DERIVATIONS.

The variational lower bound (VLB)is :

$$\mathbb{E}[-\log p_{\boldsymbol{\theta}}(\boldsymbol{z}^{(0)})] \le \underbrace{E_q\left[\log \frac{q(\boldsymbol{x}^{(1:T)}, \boldsymbol{\eta}|\boldsymbol{x}^{(0)})}{p_{\boldsymbol{\theta}}(\boldsymbol{z}^{(0:T)}, \boldsymbol{\eta}))}\right]}_{VLB} \tag{16}$$

---

**Algorithm 4** Learning algorithm of our model

---

Generating the unobserved confounders $z_1, ..., z_m$ through Algorithm 3.
Indicate the observational data $(x_1, z_1, t_1, y_1), ..., (x_m, z_m, t_m, y_m)$.
Indicate the scaling parameter $\alpha$ and $\beta$ .
Initialize all the model parameters.
Indicate the epoch number $E$.
Compute $u = \frac{1}{m} \sum_{i=1}^{m} t_i$.
Compute $w_i = \frac{t_i}{2u} + \frac{1-t_i}{2(1-u)}$ for $i = 1, ..., m$

$e = 0$ **to** $E$ Sample mini-batch data $\mathcal{B}$ from $D$ Compute the gradients of the empirical loss:

$$g_1 = \nabla_W \frac{1}{|\mathcal{B}|} \sum_{i=1}^{|\mathcal{B}|} w_i L(y_i, f(\Phi(x_i, z_i), t_i))$$

Compute the gradients of the regularization:

$$g_2 = \nabla_W \beta \mathcal{R}(f)$$

Compute the gradients of the IPM term:

$$g_3 = \nabla_W \alpha IPM_G(\hat{p}_\Phi^{t=1}, \hat{p}_\Phi^{t=0})$$

Obtain the step size scalar $\rho$ with the Adam        Update the parameters:

$$W \leftarrow W - \rho(g_1 + g_2 + g_3)$$

---

*Proof.* We present the detailed derivations of the Negative Log-Likelihood in Eq. 16.

$$
\begin{aligned}
&- \log p_{\boldsymbol{\theta}}(\boldsymbol{z}^{(0)}) \\
&\leq \underbrace{- \log p_{\boldsymbol{\theta}}(\boldsymbol{z}^{(0)}) + D_{KL}(q(\boldsymbol{x}^{(1:T)}, \boldsymbol{\eta}|\boldsymbol{x}^{(0)})||p_{\boldsymbol{\theta}}(\boldsymbol{z}^{(1:T)}|\boldsymbol{z}^{(0)}, \boldsymbol{\eta}))}_{A} \\
&\leq \log p_{\boldsymbol{\theta}}(\boldsymbol{z}^{(0)}) + \underbrace{E_q \left[ \log \frac{q(\boldsymbol{x}^{(1:T)}, \boldsymbol{\eta}|\boldsymbol{x}^{(0)})}{p_{\boldsymbol{\theta}}(\boldsymbol{z}^{(1:T)}|\boldsymbol{z}^{(0)}, \boldsymbol{\eta}))} \right]}_{B} \\
&\leq - \log p_{\boldsymbol{\theta}}(\boldsymbol{z}^{(0)}) + \underbrace{E_q \left[ \log \frac{q(\boldsymbol{x}^{(1:T)}, \boldsymbol{\eta}|\boldsymbol{x}^{(0)})}{p_{\boldsymbol{\theta}}(\boldsymbol{z}^{(0:T)}, \boldsymbol{\eta}))} \right] + \log p_{\boldsymbol{\theta}}(\boldsymbol{z}^{(0)})}_{C} \\
&\leq \underbrace{E_q \left[ \log \frac{q(\boldsymbol{x}^{(1:T)}, \boldsymbol{\eta}|\boldsymbol{x}^{(0)})}{p_{\boldsymbol{\theta}}(\boldsymbol{z}^{(0:T)}, \boldsymbol{\eta}))} \right]}_{VLB}
\end{aligned}
\tag{17}
$$

$\square$

We can further derive the $L_{VLB}$ as:

$$
\begin{aligned}
L_{VLB} &= E_q \left[ \log \frac{q(\boldsymbol{x}^{(1:T)}, \boldsymbol{\eta}|\boldsymbol{x}^{(0)})}{p_{\boldsymbol{\theta}}(\boldsymbol{z}^{(0:T)}, \boldsymbol{\eta}))} \right] \\
&= E_q \left[ \sum_{t=2}^{T} D_{KL} \left( \underbrace{q(\boldsymbol{x}^{(t-1)}|\boldsymbol{x}^{(t)}, \boldsymbol{x}^{(0)})}_{A} || \underbrace{p_{\boldsymbol{\theta}}(\boldsymbol{z}^{(t-1)}|\boldsymbol{z}^{(t)}, \boldsymbol{\eta})}_{B} \right) \right. \\
&\quad \left. - \log \underbrace{p_{\boldsymbol{\theta}}(\boldsymbol{z}^{(0)}|\boldsymbol{z}^{(1)}, \boldsymbol{\eta})}_{C} + D_{KL} \left( \underbrace{q_{\boldsymbol{\varphi}}(\boldsymbol{\eta}|\boldsymbol{x}^{(0)})}_{D} || \underbrace{p(\boldsymbol{\eta})}_{E} \right) \right]
\end{aligned}
\tag{18}
$$

*Proof.* We present the detailed derivations of the VLB in Eq. 18.

$$L_{VLB} = E_q \left[ \log \frac{q(\boldsymbol{x}^{(1:T)}, \boldsymbol{\eta}|\boldsymbol{x}^{(0)})}{p_{\boldsymbol{\theta}}(\boldsymbol{z}^{(0:T)}, \boldsymbol{\eta}))} \right]$$

$$= E_q \left[ \log \frac{q(\boldsymbol{\eta}|\boldsymbol{x}^{(0)}) \prod_{t=1}^{T} q(\boldsymbol{x}^{(t)}|\boldsymbol{x}^{(t-1)})}{p_{\boldsymbol{\theta}}(\boldsymbol{\eta})p(\boldsymbol{z}^{(T)}) \prod_{t=1}^{T} p_{\boldsymbol{\theta}}(\boldsymbol{z}^{(t-1)}|\boldsymbol{z}^{(t)}, \boldsymbol{\eta})} \right]$$

$$= E_q \left[ -\log p(\boldsymbol{z}^{(T)}) + \sum_{t=1}^{T} \log \frac{q(\boldsymbol{x}^{(t)}|\boldsymbol{x}^{(t-1)})}{p_{\boldsymbol{\theta}}(\boldsymbol{z}^{(t-1)}|\boldsymbol{z}^{(t)}, \boldsymbol{\eta})} + \log \frac{q_{\varphi}(\boldsymbol{\eta}|\boldsymbol{x}^{(0)})}{p_{\boldsymbol{\theta}}(\boldsymbol{\eta})} \right]$$

$$= E_q \left[ -\log p(\boldsymbol{z}^{(T)}) + \log \frac{q(\boldsymbol{x}^{(1)})|\boldsymbol{x}^{(0)})}{p_{\boldsymbol{\theta}}(\boldsymbol{z}^{(0)}|\boldsymbol{z}^{(1)}), \boldsymbol{\eta})} + \sum_{t=2}^{T} \log \left( \frac{q(\boldsymbol{x}^{(t-1)}|\boldsymbol{x}^{(t)}, \boldsymbol{x}^{(0)})}{p_{\boldsymbol{\theta}}(\boldsymbol{z}^{(t-1)}|\boldsymbol{z}^{(t)}, \boldsymbol{\eta})} \cdot \frac{q(\boldsymbol{x}^{(t)}|\boldsymbol{x}^{(0)})}{q(\boldsymbol{x}^{(t-1)}|\boldsymbol{x}^{(0)})} \right) + \log \frac{q_{\varphi}(\boldsymbol{\eta}|\boldsymbol{x}^{(0)})}{p_{\boldsymbol{\theta}}(\boldsymbol{\eta})} \right]$$

$$= E_q \left[ -\log p(\boldsymbol{z}^{(T)}) + \log \frac{q(\boldsymbol{x}^{(1)})|\boldsymbol{x}^{(0)})}{p_{\boldsymbol{\theta}}(\boldsymbol{z}^{(0)}|\boldsymbol{z}^{(1)}), \boldsymbol{\eta})} + \sum_{t=2}^{T} \log \frac{q(\boldsymbol{x}^{(t-1)}|\boldsymbol{x}^{(t)}, \boldsymbol{x}^{(0)})}{p_{\boldsymbol{\theta}}(\boldsymbol{z}^{(t-1)}|\boldsymbol{z}^{(t)}, \boldsymbol{\eta})} + \log \frac{q(\boldsymbol{x}^{(T)}|\boldsymbol{x}^{(0)})}{q(\boldsymbol{x}^{(1)}|\boldsymbol{x}^{(0)})} + \log \frac{q_{\varphi}(\boldsymbol{\eta}|\boldsymbol{x}^{(0)})}{p_{\boldsymbol{\theta}}(\boldsymbol{\eta})} \right]$$

$$= E_q \left[ -\log \frac{p(\boldsymbol{x}^{(T)})}{q(\boldsymbol{x}^{(T)}|\boldsymbol{x}^{(0)})} - \log p_{\boldsymbol{\theta}}(\boldsymbol{z}^{(0)}|\boldsymbol{z}^{(1)}), \boldsymbol{\eta}) + \sum_{t=2}^{T} \log \frac{q(\boldsymbol{x}^{(t-1)}|\boldsymbol{x}^{(t)}, \boldsymbol{x}^{(0)})}{p_{\boldsymbol{\theta}}(\boldsymbol{z}^{(t-1)}|\boldsymbol{z}^{(t)}, \boldsymbol{\eta})} + \log \frac{q_{\varphi}(\boldsymbol{\eta}|\boldsymbol{x}^{(0)})}{p_{\boldsymbol{\theta}}(\boldsymbol{\eta})} \right]$$

$$= E_q \left[ \sum_{t=2}^{T} D_{KL} \left( q(\boldsymbol{x}^{(t-1)}|\boldsymbol{x}^{(t)}, \boldsymbol{x}^{(0)})||p_{\boldsymbol{\theta}}(\boldsymbol{z}^{(t-1)}|\boldsymbol{z}^{(t)}, \boldsymbol{\eta}) \right) - \log p_{\boldsymbol{\theta}}(\boldsymbol{z}^{(0)}|\boldsymbol{z}^{(1)}, \boldsymbol{\eta}) + D_{KL} \left( q_{\varphi}(\boldsymbol{\eta}|\boldsymbol{x}^{(0)})||p_{\boldsymbol{\theta}}(\boldsymbol{\eta}) \right) \right]$$

$$\tag{19}$$

$\square$

# F    PSEUDO-CODE OF DFHTE

We present the diffusion model training algorithm in Algorithm 2, the sampling algorithm in Algorithm 3, and our CATE estimation algorithm in Algorithm 4.

