# OpenReview forum: "Mitigating Unobserved Confounding via Diffusion Probabilistic Models"
_ICLR.cc/2025/Conference — Submitted to ICLR 2025_

### Official Review · Reviewer_wpss · 2024-10-24

**Soundness:** 1
**Presentation:** 2
**Contribution:** 1
**Rating:** 3
**Confidence:** 5

**Summary:**

This paper aims to estimate the CATE without Ignorability assumption. It proposes a method to capture the unobserved latent space (generate the unobserved confounders) using diffusion models and then use it to estimate the causal effect.

**Strengths:**

Causal inference under hidden confounding is an important problem.

**Weaknesses:**

## Main weakness
1. This paper tackles the challenging problem of estimating the CATE under unobserved confounding. However, without additional assumptions (e.g., instruments), it is impossible to get identification of CATE or unbiased estimation, regardless of the latent variable model used. Therefore, it is impossible to solve the task of unobserved confounders in causal inference with the diffusion model as the paper claimed for its contribution.

Current research in causal inference discusses partial identification and sensitivity analysis with unobserved confounding settings, e.g. [A Neural Framework for Generalized Causal Sensitivity Analysis]. Authors should explore these directions more.


## Other weakness
2. The paper lacks a thorough literature review on using diffusion models for CATE and counterfactual estimation. Specifically, it fails to cite and differentiate itself from closely related works, such as [Interventional and Counterfactual Inference with Diffusion Models] and [Diffusion Causal Models for Counterfactual Estimation]. Despite the identifiability of the proposed method in this paper, it does not adequately discuss how its approach compares to other existing methods which also use diffusion models for the same task.

3. The variational lower bound presented is trivial, essentially involving a simple replacing notation from standard diffusion models. Not necessarily counts as a contribution.

4. In terms of experimental results, the proposed method performs similarly to TARNet, and in some cases, even worse.

In summary, the paper does not address the issue of identifiability as it claims and also lacks a literature review on using diffusion models for CATE estimation.

**Questions:**

What is the point of showing Figure 4 and Figure 5? I didn't get this visualization of the balanced representations here meaning what and how it contributes to the overall argument.

---

### Official Review · Reviewer_J14T · 2024-11-02

**Soundness:** 2
**Presentation:** 3
**Contribution:** 2
**Rating:** 3
**Confidence:** 4

**Summary:**

The paper proposes a diffusion model-based method for causal effect estimation without the unconfoundedness assumption. Specifically, the proposed method approximates the generation factor $\eta$ which affects $x$ and $z$ first and then uses $\eta$ to recover latent confounders $z$ via diffusion model. The experimental results show the effectiveness of the proposed method.

**Strengths:**

The paper focuses on an important problem of causal effect estimation under latent confounding.

**Weaknesses:**

Weaknesses / Questions

1. Could you provide more discussion on $\eta$, including its physical meaning and how it works for recovering $z$? And it is unclear why we can learn $p(\eta|x)$ correctly. If so, why can't we directly learn $p(z|x)$ correctly as $z,\eta$ are all the direct parents of $x$​ in the assumed causal graph?
2. My biggest concern is the lack of identifiability of $z$​​ (e.g., iVAE [1]), which means that the proposed method can not guarantee the correctness of learned representations. Additionally, it could be better to provide the MCC results ($\hat z$ and $z$).
3. It could be better to provide more discussion on the experimental results.
   1. Since ACIC and IHDP datasets contain all observed confounders, why can the proposed method outperform existing methods?
   2. In the Sim-z dataset, $x$ and $z$ are independent (which does not follow the assumed causal graph), is it possible to recover $z$ by using $x$?
   3. The only dataset following the assumed causal graph is the Sim-$\eta$, however, it seems that it only achieves a similar result to TARNet.
4. It could be better to compare with recent methods, e.g., [2,3,4].



Minors (that do not influence my score)

Line 87: likelihoodof -> likelihood of



[1] Khemakhem I, Kingma D, Monti R, et al. Variational autoencoders and nonlinear ica: A unifying framework[C]//International conference on artificial intelligence and statistics. PMLR, 2020: 2207-2217.

[2] Ma Y, Melnychuk V, Schweisthal J, et al. DiffPO: A causal diffusion model for learning distributions of potential outcomes[J]. arXiv preprint arXiv:2410.08924, 2024.

[3] Zhang W, Liu L, Li J. Treatment effect estimation with disentangled latent factors[C]//Proceedings of the AAAI Conference on Artificial Intelligence. 2021, 35(12): 10923-10930.

[4] Guo X, Zhang Y, Wang J, et al. Estimating heterogeneous treatment effects: Mutual information bounds and learning algorithms[C]//International Conference on Machine Learning. PMLR, 2023: 12108-12121.

**Questions:**

see weaknesses

---

### Official Review · Reviewer_FLzA · 2024-11-04

**Soundness:** 2
**Presentation:** 3
**Contribution:** 2
**Rating:** 5
**Confidence:** 4

**Summary:**

The paper addresses the challenge of estimating CATE from observational data in the presence of unobserved confounders. The authors propose an approach using a diffusion model to generate unobserved confounders and thus enable the estimation of causal effects. This method employs both forward and reverse diffusion processes to effectively reconstruct hidden confounders. They derive a variational lower bound for the likelihood of these unobserved confounders.

**Strengths:**

- The incorporation of a diffusion model could be an advantage in recovering unobserved confounders. However, I believe that the authors should clearly state the assumptions under which the causal effect is identifiable.

- The proposed framework shows competitive performance on some datasets. However, there are some concerns on the experimental results as outlined below.

**Weaknesses:**

1/ It appears to me that the proposed method is a straightforward application of a diffusion model, used to infer unobserved confounders. I believe it can be directly applied to methods that address unobserved confounders, such as CEVAE.

2/ The inference of unobserved confounders is based on variational inference, which could be unidentifiable unless further assumptions are made. This is known as posterior collapse and is well documented in:

- Wang, Y., Blei, D., & Cunningham, J. P. (2021). Posterior collapse and latent variable non-identifiability. Advances in neural information processing systems, 34, 5443-5455.

Can the authors show that the latent confounders are identifiable with the diffusion model?

In general, I believe that the causal effects in this case is unidentifiable unless the authors limit their work with specific assumptions as discussed in CEVAE:

- Louizos, C., Shalit, U., Mooij, J. M., Sontag, D., Zemel, R., & Welling, M. (2017). Causal effect inference with deep latent-variable models. Advances in neural information processing systems, 30.

3/ The causal quantity of interest in this paper is CATE, defined in Section 2.1. This quantity is only conditional on the observed confounder $x$. To estimate this quantity, we need to block the backdoor through $z$. It is unclear how they blocked it. Did you sample $z$ from the diffusion model and use these samples to block the backdoor? In that case, the concern goes back to point 2/ above -- we cannot recover the latent confounders due to posterior collapse.

4/ Since the unobserved confounder $z$ is learned from the observed confounder $x$, this could be a big problem if they are 'independent'. How can you infer $z$ from a variable $x$ which are independent of it?

5/ Concerns regarding the datasets used in experiments:
- IHDP data is randomised and all confounders are observed. Why do we need to infer unobserved confounder?

- From the simulation of Sim-z (Eq. 12), $z$ and $x$ are independent. It means that you cannot infer $z$ from $x$ since there is no correlation/causal relationship between them. This concern is related to point 4/ above.

This raises a significant concern regarding the experimental comparison on these two datasets.

6/ In Eq. 8, $\Phi(\cdot)$ is a function of the observed confounder $x$. However, in Eq. 10, it is a function of $x$ and $z$. Can the author(s) explain why there are such differences? Isn't the input to $\Phi(\cdot)$ should be a fixed vector?

7/ Table 1 should be Figure 1?

**Questions:**

Please refer to the questions in the Weaknesses section.

---

### Official Review · Reviewer_WAFy · 2024-11-11

**Soundness:** 2
**Presentation:** 2
**Contribution:** 2
**Rating:** 6
**Confidence:** 3

**Summary:**

The paper proposes a novel approach to estimating CATE from observational data, addressing the unobserved confounder challenge in causal inference. The model estimates causal effects by modeling unobserved confounders via a reverse diffusion process, formulated as a Markov chain conditioned on prior knowledge. Empirical results on synthetic and real-world data suggest consistent improvements over state-of-the-art methods.

**Strengths:**

1. This paper proposes a novel approach by leveraging a diffusion model to capture unobserved confounders in causal inference.
2. The authors validate their model on both synthetic and real-world datasets, showing consistent improvements over baseline methods, which suggests robust performance across varied settings.

**Weaknesses:**

The computational cost of the diffusion process itself might be a concern, especially for large-scale real-world datasets. Further discussion on the computational complexity and potential bottlenecks of the diffusion process would be beneficial for understanding its practical applicability.

**Questions:**

See above.

---

### Meta-Review · Area_Chair_wsWE · 2024-12-07

**Metareview:**

This paper tackles the problem of estimating the Conditional Average Treatment Effect (CATE) from observational data when unobserved confounders are present. It introduces a diffusion model capable of generating these hidden confounders, thus facilitating causal effect estimation. Its proposed method leverages both forward and reverse diffusion processes to accurately reconstruct the unobserved variables, and they further establish a variational lower bound on the likelihood of these latent confounders.

The reviewers have several critical concerns regarding this paper. Notably, there are concerns regarding (1) lack of discussion around the issue of identification in causal inference; (2) lack of thorough literature review; (3) incremental technical contribution (i.e., straight-forward application of diffusion models); (4) insignificant empirical results in some cases (e.g., comparable or worse than TARnet).

However, there is no rebuttal. All concerns raised by the reviewers have therefore not been addressed.
Given this, I recommend a rejection.

**Additional Comments On Reviewer Discussion:**

There is no discussion in this case because the authors have not provided a rebuttal to address the concerns raised by reviewers.

---

### Decision · Program_Chairs · 2025-01-22

Reject